# Inhibition of Calpain Attenuates Degeneration of Substantia Nigra Neurons in the Rotenone Rat Model of Parkinson’s Disease

**DOI:** 10.3390/ijms232213849

**Published:** 2022-11-10

**Authors:** Vandana Zaman, Kelsey P. Drasites, Ali Myatich, Ramsha Shams, Donald C. Shields, Denise Matzelle, Azizul Haque, Narendra L. Banik

**Affiliations:** 1Ralph H. Johnson Veterans Administration Medical Center, 109 Bee St., Charleston, SC 29401, USA; 2Department of Neurosurgery, Medical University of South Carolina, 96 Jonathan Lucas St., Charleston, SC 29425, USA; 3The Citadel, 171 Moultrie St., Charleston, SC 29409, USA; 4Department of Microbiology and Immunology, Medical University of South Carolina, 173 Ashley Avenue, Charleston, SC 29425, USA

**Keywords:** calpain, dopaminergic neuron, alpha-synuclein, microglia, Parkinson’s disease, rotenone, substantia nigra

## Abstract

In the central nervous system (CNS), calcium homeostasis is a critical determinant of neuronal survival. Calpain, a calcium-dependent neutral protease, is widely expressed in the brain, including substantia nigra (SN) dopaminergic (DA) neurons. Though calpain is implicated in human Parkinson’s disease (PD) and corresponding animal models, the roles of specific ubiquitous calpain isoforms in PD, calpain-1 and calpain-2, remain poorly understood. In this study, we found that both isoforms are activated in a nigrostriatal pathway with increased phosphorylated synuclein following the administration of rotenone in Lewis rats, but calpain isoforms played different roles in neuronal survival. Although increased expression of calpain-1 and calpain-2 were detected in the SN of rotenone-administered rats, calpain-1 expression was not altered significantly after treatment with calpain inhibitor (calpeptin); this correlated with neuronal survival. By contrast, increased calpain-2 expression in the SN of rotenone rats correlated with neuronal death, and calpeptin treatment significantly attenuated calpain-2 and neuronal death. Calpain inhibition by calpeptin prevented glial (astroglia/microglia) activation in rotenone-treated rats in vivo, promoted M2-type microglia, and protected neurons. These data suggest that enhanced expression of calpain-1 and calpain-2 in PD models differentially affects glial activation and neuronal survival; thus, the attenuation of calpain-2 may be important in reducing SN neuronal loss in PD.

## 1. Introduction

Parkinson’s disease (PD) is a common neurodegenerative disorder with characteristic symptoms including resting tremor, bradykinesia, gait instability, and rigidity. The debilitating motor symptoms are due to progressive loss of midbrain dopaminergic (DA) substantia nigra (SN) neurons. These cells often demonstrate abnormal cytoplasmic aggregations of α-synuclein (α-syn) known as Lewy bodies [1,2]. Though various cellular and molecular events play a critical role in the onset and progression of this neurodegenerative disease [2], the etiopathology of neuronal loss in sporadic PD cases is still incompletely understood.

Less than 40% of SN DA neurons express the calcium-binding proteins calbindin and/or calretinin [3,4]. The presence of these calcium-binding proteins helps provide resistance to toxin-induced cell death [5], suggesting that intracellular calcium homeostasis is a critical determinant of neuronal survival [6]. Calpain, a calcium-dependent neutral protease, [7], is widely expressed in the brain (including SN DA neurons [8]), and tyrosine hydroxylase (TH), a rate-limiting enzyme involved in dopamine synthesis, is a calpain substrate. Since calpain activity influences TH function, pathological persistent hyper-activation of calpain may damage the neurons [6,9]. Post-mortem analyses of brains from PD patients reveal increased levels of calpain-2 in the SN neurons [10]. PD animal model findings also support calpain involvement in Parkinsonian pathogenesis [11]; likewise, inhibition of its activity may provide neuroprotection.

Another calpain substrate with a significant role in PD pathogenesis is α-syn [9,12]. In vitro studies have demonstrated that calpain cleaves the C-terminal region of fibrillated α-syn. The fragments generated due to this cleavage promote α-syn aggregation and may form Lewy bodies [12,13]. Moreover, these α-syn fragments induce oxidative stress in DA neurons, leading to neuronal death [14]. The aggregated form of α-syn is a predominant neurotoxic component of Lewy bodies in PD [6]. Additionally, α-syn plays a key role in dopamine biosynthesis, and the mutant form of α-syn increases TH activity [15,16]. Furthermore, the regulation of TH by aggregated α-syn and/or calpain activation may lead to dopamine overproduction in nigral neurons, eventually causing neuronal degeneration. 

PD animal model studies have provided valuable information regarding various pathways regulating the maintenance of normal neuronal functions and specific subtle changes in these pathways that lead to neurodegeneration [2]. Betarbet et al. demonstrated that systemic administration of rotenone in nonhuman primates inhibits mitochondrial complex I and leads to selective loss of SN DA neurons [17]. This model also replicates crucial histopathological features of clinical PD: α-syn phosphorylation, α-syn aggregation, and Lewy body formation [18,19]. These pathological findings also found in rodents following rotenone administration make this model popular for the investigation of PD pathogenesis [17,19,20,21,22]. 

We have previously shown that early intervention by a calpain inhibitor, calpeptin, protects motor neurons in a mouse model of PD [23,24,25]. In vitro studies showed that rotenone treatment elevated intracellular free Ca^2+^ and calpain in SH-SY5Y cells, and calpain inhibition protected cell viability and preserved cellular morphology following rotenone exposure [26]. In the present study, we used the rotenone rat model to investigate the role of two ubiquitous calpain isoforms, calpain-1 and calpain-2, in the neurodegenerative processes to determine whether calpain inhibition attenuates midbrain DA neuronal death and improves symptomatic outcomes. A pan-calpain inhibitor, calpeptin, was used as a neuroprotective strategy to inhibit glial activation and prevent the loss of SN DA neurons. Our immunohistochemical and Western blot data demonstrated increased expression of calpain-1 and calpain-2 in the SN DA neurons after rotenone administration in Lewis rats. Moreover, rotenone administration led to glial activation and neuroinflammation in the nigrostriatal pathway, and calpain inhibition promoted microglial M2 differentiation and prevented the loss of SN neurons. These data suggest that the activation of calpain isoforms by rotenone may have opposing roles in neuroinflammation and DA neuron degeneration.

## 2. Results

### 2.1. Increased Expression of Phosphorylated α-syn and Induction of Gliosis in Dorsal Striatum of Rotenone-Parkinsonian Rats

The aggregation of α-syn is thought to play a significant role in the onset of PD, whereby α-syn is abnormally phosphorylated on Ser-129 [27]. Western blot analysis of lysates from the dorsal striatum of Lewis rats injected with rotenone showed an increased expression of p-α-syn129 proteins (Figure 1A). The expression level of this p-α-syn protein was significantly higher (*p* < 0.0001) in rotenone-injected rats as compared to vehicle controls (Figure 1B). In pathological situations following rotenone injection, astrocytes can be activated and may produce inflammatory factors, leading to neurodegeneration in PD [28]. Immunostaining of the glial fibrillary acidic protein (GFAP), a marker for astrocytes, showed that astrocytes were localized around blood vessels in the SN (Figure 1C). These glial cells were in small clusters around blood vessels in the control group in which astrocytes appeared normal in size with thin, long processes. By contrast, astrocytes were densely distributed evenly in the SN of rotenone-administered groups, and the phenotype was altered. Quantitative analysis of GFAP+ cells suggested that the number of astrocytes was increased with hypertrophied, highly branched processes (arrows in Figure 1C) as compared to vehicle controls (Figure 1D). These data suggest that the administration of rotenone induced parkinsonism with increased phosphorylation of α-syn129 and active gliosis in Lewis rats.

### 2.2. Inhibition of Calpain Prevents Rotenone-Induced Loss of DA Neurons in the Substantia Nigra of Rotenone-Parkinsonian Rats

Though calcium homeostasis is important for the maintenance of neuronal integrity, elevated calcium induces calpain activation and the aggregation of α-syn, as well as the stimulation of gliosis in PD. To investigate whether calpain inhibition by calpeptin (CP) prevents rotenone-induced SN neuronal loss, TH immunofluorescence staining was performed (Figure 2A). The TH neuronal loss in SN pars compacta (SNpc, arrows) was evident following rotenone administration as compared to the untreated controls. Along with neuronal loss, reduced DA fiber density was also distinct in SN pars reticulata (SNpr, arrowheads) following rotenone injection. Interestingly, the neuroprotective effects of calpeptin were evident in SN of the rotenone+CP treatment group, as TH-stained neurons and fibers were very prominent in the SNpc (arrows) and SNpr (arrowheads), respectively. Calpain inhibition significantly blocked TH neuronal loss in SNpc and fibers in SNpr (Figure 2B), as shown in the quantitative image analysis of TH-positive fiber density in Figure 2B. These data suggest that calpain inhibition attenuates rotenone-induced loss of DA neurons in the substantia nigra and may prevent neurodegeneration. 

### 2.3. Increased Expression of Calpain-1 Was Detected in Substantia Nigra DA Neurons of Rotenone-Parkinsonian Rats

Although both calpain-1 and calpain-2 are reported to be activated in neurodegenerative diseases, they may play opposite roles in cell survival and death [29]. Calpain-1 immunofluorescence staining in the SN of rotenone rats showed precise localization of this protease in SN neurons and fibers in the control rats (Figure 3A). The co-localization of calpain-1 with TH suggests that calpain-1-expressing neurons are SN DA neurons. Figure 3A also provides representative images from the respective treatment groups. The SN DA neurons demonstrated cytoplasmic expression of calpain-1 in the control rats. This co-localization also delineates a distinct nucleus (counterstained with DAPI) with the rim of the intense TH expression (green) compared to calpain-1 expression. The TH-immunostained DA fibers also demonstrate calpain-1 (co-localization). It appears that rotenone administration increased calpain-1 expression in the DA SN neurons. The co-localization with TH also shows the intensely stained (yellow-orange) SN neurons, further suggesting up-regulated expression of calpain-1 following rotenone administration. Moreover, this enhanced expression of calpain-1 in DA fibers is also distinct. Immunostaining of SN from the rotenone+CP treatment group indicates that some SN neurons demonstrate enhanced calpain-1 expression compared to controls. Quantitative analyses of colocalized TH and calpain-1 suggest that calpain-1 is significantly increased in neurons after rotenone injection, and it was marginally inhibited by CP treatment (Figure 3B). These data suggest that calpain-1 may support neuronal survival. 

### 2.4. Calpeptin Treatment Reduced the Enhanced Expression of Calpain-2 in SN DA Neurons of Rotenone-Parkinsonian Rats

Like calpain-1, we analyzed calpain-2 expression via immunostaining in SN DA neurons. Similar to calpain-1, calpain-2 expression was also detected in SN neurons, and the specific co-localization with TH suggests that these were dopaminergic neurons (Figure 4A). Immunostaining and the co-localization of TH and calpain-2 showed that rotenone administration promoted the expression of calpain-2 in nigral DA neurons. However, in the rotenone+CP treatment group, most of the SN neurons expressed significantly more TH than calpain-2 (Figure 4A,B). Quantitative analysis of colocalized cells (Figure 4B) suggested that neuronal calpain-2 expression was attenuated following CP treatment and might have supported neuronal survival. Thus, calpain-1 and calpain-2 isoforms could play opposing functions after CP treatment in rotenone-injected rats. 

### 2.5. Calpeptin Treatment Attenuated Reactive Gliosis in the Dorsal Striatum of Rotenone-Parkinsonian Rats

Astrocytes possess a calcium-based form of excitability. As mentioned earlier (Figure 1), the number of astrocytes was increased with hypertrophied, highly branched processes following rotenone administration. These reactive astrocytes contribute to the degenerative process and/or repair mechanisms in the CNS. Immunostaining of astrocytes (GFAP) demonstrated significantly more astrocytes in the dorsal striatum following rotenone injection as compared to the vehicle control (Figure 5A,B, *p* < 0.0131). These activated astrocytes appeared to have many long, branched processes (arrows), and they were significantly inhibited by CP treatment (Figure 5B, *p* = 0.0314). Immunohistochemical analysis also showed significant microglial (Iba1) proliferation in the SN of rotenone rats as compared to vehicle controls (Figure 5B,C; *p* = 0.0105). Treatment of rats with CP failed to significantly decrease microglial numbers in the SN (Figure 5D, *p* = 0.0535), although there was a trend toward decreased numbers. We then tested whether microglia underwent differentiation into the M2 phenotype following calpain inhibition. Immunostaining of the dorsal striatum with Iba1 and Arginase 1 antibodies showed the presence of increased Iba-1^+-^/Arginase 1^+^-positive cells in CP-treated rats (arrows) (Figure 5E). Quantitative analyses of colocalized Iba1- and Arginase 1-positive cells (Figure 5F) suggest calpain inhibition may have promoted the differentiation of M2-type microglia.

### 2.6. Calpain Inhibitor Attenuates Rotenone-Induced Calpain-2 in Dorsal Striatum of Rotenone-Parkinsonian Rats

Though calpain-1 and calpain-2 are predominantly expressed in mammalian brain, calpain-2 is implicated in neurodegeneration [30]. Western blot analysis showed a significant increase in calpain-1 (*p* < 0.0005) and calpain-2 (*p* < 0.0001) expression in the dorsal striatum of rotenone-injected rats as compared to vehicle controls (Figure 6). Calpeptin treatment marginally decreased calpain-1 expression; this was not significant (*p* = 0.1171) (Figure 6A). However, treatment of rotenone rats with CP (calpeptin) significantly inhibited (*p* < 0.0001) calpain-2 expression in the striatum, as analyzed by Western blotting and ImageJ software (Figure 6B). Interestingly, CP treatment reduced the expression of rotenone-induced calpain-2 to almost the level observed in controls. These data suggest the importance of calpain-2 in Parkinsonian neurodegeneration, and the attenuation of calpain-2 may improve neuronal survival in rotenone-induced PD models.

## 3. Discussion 

Calpain-1 and calpain-2 are the major calpain isoforms in the brain, and they may play opposite roles in neuroprotection versus neurodegeneration [31]. Calpain is implicated in the pathogenesis of many neurological conditions such as epilepsy, stroke, spinal cord injury, and traumatic brain injury (TBI) [6,25,31,32,33,34,35,36,37,38,39,40,41]. Our present study demonstrated: (i) Rotenone administration induced upregulation of both calpain-1 and calpain-2 expression and phosphorylated α-syn in nigrostriatal dopaminergic pathway. (ii) The inhibition of calpain by calpeptin significantly decreased calpain-2 and glial activation, promoted M2 microglial differentiation, and protected DA neurons in rotenone-injected rats. In accordance with other studies [10,23,37,42], enhanced expression of calpains in the nigrostriatal pathway suggests that these calcium-dependent proteases may play critical roles in the pathogenesis of PD.

Intracellular Ca^2+^ homeostasis is maintained for normal DA cellular processes. Elevated Ca^2+^ levels thus lead to persistent aberrant calpain activity, potentially contributing to acute and chronic neurodegeneration [7]. Our previous studies have reported the active role of calpains in PD, spinal cord injury, and optic neuritis [23,25,38,43,44,45,46,47,48]. As reported by other groups [42,49], we also detected calpain activation in DA neurons in rotenone-injected rats. We found increased expression of calpain-1 and calpain-2 in the SN DA neurons of rats following rotenone administration. Our data from Western blot analysis also demonstrated significantly increased calpain-1 and calpain-2 protein expression in the dorsal striatum of the rotenone-administered group. These findings suggest that rotenone administration may lead to upregulated calpain-1 and -2 expression in the nigrostriatal DA pathway. Surprisingly, the expression of calpain-1 was not significantly downregulated by calpeptin treatment in rotenone rats. However, calpain-2 levels were significantly downregulated by calpeptin treatment, suggesting differential roles of calpain-1 and calpain-2 in the neurodegenerative process of rotenone-induced PD in rats. Studies using calpain-1 knockout and calpain-2 conditional knockout mice have shown that calpain-1 could be neuroprotective, and calpain-2 may lead to neuronal death or degeneration [50,51]. Our present study suggests that the expression of calpain-1 remains unchanged or marginally reduced following pan-calpain inhibitor treatment in rotenone rats. However, calpain-2 was significantly inhibited, which correlated with the improved outcome in rotenone rats, suggesting that calpain-2 inhibition could be neuroprotective. Thus, calpain-1 and calpain-2 might play opposing roles in neurodegenerative diseases. Wang et al. also demonstrated that prolonged activation of calpain-2 is critical in inducing pathological changes in the hippocampus following seizures [50], suggesting that calpain-2 activity may promote neuronal death, whereas calpain-1 supports neuronal survival [29]. Our study, however, found enhanced expression of both calpain isoforms in SN DA neurons, suggesting calpain expression may be related to SN neuroinflammation rather than local DA cell death [52]. Our study also detected the sustained enhanced expression of both calpain isoforms at one-month post-injection of rotenone. Thus, the interplay between calpain-1 and calpain-2 may influence the fate of these nigral neurons in chronic conditions, as recently reported [53]. 

Mitochondrial dysfunction is thought to be one of the critical factors in the onset and progression of PD [2], and several studies have detected reduced complex I activity in PD patients [54,55]. Rotenone is a potent inhibitor of complex I [17,56]. The specific loss of SN neurons caused by rotenone’s systemic toxicity suggests that the nigrostriatal pathway is intrinsically and selectively vulnerable to complex I inhibition [57]. Previously, we observed increased calpain activity and inflammation in the spinal cord of PD patients [58] with caspase -3 activation following rotenone injection [25]. 

Calpain activation is likewise detrimental to cell survival, because it stimulates apoptotic pathways [59]. Calpain also inactivates the complex of Beclin, an autophagy gene—thus promoting apoptosis rather than autophagy [60,61]. Autophagy plays a critical role in clearing damaged intracellular organelles to maintain cellular homeostasis. Blocking this autophagy process may lead to the accumulation of damaged cellular organelles and misfolded proteins, causing cell death via apoptosis [2]. Disruption in the autophagy pathway may cause an increased level of phosphorylated α-syn 129 in the rotenone-induced PD model. Additionally, blocking calpain-2 activation by calpeptin treatment might activate the autophagy pathway, protecting neurons from the toxic aggregation of α-syn [62]. 

Disruption of calcium homeostasis in neurodegenerative processes may lead to calpain activation [33]. SN DA neurons expressing calbindin and calretinin are found to be resistant to neurodegeneration, suggesting that calcium-binding proteins are neuroprotective for SN DA neurons [5,63,64]. The immunohistochemical localization of both calpain-1 and calpain-2 in mid-brain SN DA neurons suggests that unregulated changes in physiological calpain activity can be detrimental to DA neurons. Calpain-2 hyperactivity was detected in synaptosomes of Alzheimer disease (AD) patients during pre-symptomatic phases [65]. Furthermore, this activation was linked to increased levels of β-amyloid deposits and a decline in cognitive functions. Although our present study demonstrated enhanced expression of both calpain-1 and calpain-2 in the SN DA neurons following rotenone administration, increased calpain-1 expression correlated with DA neuronal survival. The enhanced expression of calpain-1 in SN DA neurons could be functioning similar to calbindin and potentially protected these neurons from rotenone-induced cytotoxicity. Our study suggests that rotenone administration induced an increase in calpain-2 expression in the SN, while whereas treatment prevented hyperactivation of calpain-2. 

PD animal model studies as well as human PD brain samples have demonstrated the presence of neuroinflammation in the SN [66,67]. Microglia-mediated neuroinflammation plays a critical role in neurodegenerative diseases including PD [67]. Microglial distribution and morphology are heterogeneous in the brain, and the SN is highly populated by microglia [68,69,70], suggesting that SN DA neurons are vulnerable to neuroinflammation. In the present study, alterations of glial phenotypes indicate that rotenone administration led to the activation of microglia/astrocytes with increased glial density in the nigrostriatal pathway. Additional immunostaining identified Iba1^+^/arginase 1^+^, microglia, a marker for neuroprotective type (M2) microglia, in the nigrostriatal pathway following calpain inhibition. The absence of microglial activation with calpeptin treatment of rotenone rats strongly suggests that calpain inhibition attenuates neuroinflammation in the SN. 

These findings indicate that rotenone-induced calpain-1 and calpain-2 overexpression plays a critical role in the degeneration of SN neurons. Moreover, the inhibition of calpain overexpression prevents the extensive loss of SN DA neurons caused by rotenone administration. Microglial activation in SN caused by rotenone toxicity may also play a prominent role in neuronal death (Figure 7). Furthermore, calpain inhibition may attenuate microglial activation—specifically in the SN, but not in the striatum as part of a regionally specific microglial activation pathway (Figure 7). Thus, it is critical to investigate the relationship between calpain-1 and calpain-2 expression in SN DA neurons in PD pathogenesis. Overall, our study suggests that the inhibition of calpain, especially calpain-2, attenuates glial activation, promotes microglial differentiation into M2 cells, prevents neurodegeneration, and rescues neurons in rotenone parkinsonism. 

## 4. Material and Methods

### 4.1. Animals

Male Lewis rats (3–4 months, 300–350 g body weight, ENVIGO) were housed in an animal facility under standard conditions (12 h light-dark cycles, 23 °C, and 55% relative humidity) with ad libitum access to food and water. Rats were handled and cared for in compliance with the guidelines of the National Institutes of Health (NIH, Bethesda, MD, USA) *Guide for the Care and Use of Laboratory Animals* (NIH publication 80-23, revised 1996) and approved (ACORP 643) by the Institutional Animal Care and Use Committee (IACUC) of the Ralph H. Johnson Veteran Medical Center of South Carolina, Charleston, SC, USA. 

### 4.2. Rotenone Administration

Rats were divided into four treatment groups: (1) control+ vehicle, (2) calpeptin, (3) rotenone, and (4) rotenone plus calpeptin. Rotenone (Sigma, St. Louis, MO, USA) was injected subcutaneously (s.c.) at a dose of 2 mg/kg body weight. Groups 3 and 4 received rotenone s.c. daily for four days, and then every other day for 6 days. Group 4 also received calpeptin (Sigma, St. Louis, MO, USA) intraperitoneally (i.p.) daily at a dose of 25 µg/kg body weight. The calpeptin injections started one day after the 1st injection of rotenone (nine injections total) and were injected 1 h after rotenone injection. 

Stock solution of rotenone was prepared in dimethylsulfoxide (DMSO, Sigma). The working emulsified dilution was made in sunflower oil. The dose was prepared fresh every other day and stored in amber-colored glass vials. The calpeptin stock solution was prepared in DMSO, and working dilution was prepared in sterile saline. The working dilution was prepared fresh each time before use. The animal’s body weight was measured before starting the treatments and at the end of the treatments. 

### 4.3. Tissue Processing

One month after the last injection (20 µm thick), rats were sacrificed according to the approved protocol. Deeply anesthetized (ketamine plus xylazine) rats were decapitated, and blood was collected in EDTA-coated glass tubes (BD Vacutainer™, Thermofisher Scientific, Waltham, MA, USA). Organs were dissected and fixed in 4% paraformaldehyde or stored in dry ice before transferring to −80 °C [71]. The brain was sliced in half at the midsagittal plane, and the left hemisphere of the brain was immediately immersed in 4% paraformaldehyde for 48 h at 4 °C for analysis by immunohistochemistry. From the right hemisphere, different areas of the brain, such as the frontal cortex, striatum, and substantia nigra, were dissected and stored immediately in dry ice before transferring to −80 °C. Paraformaldehyde-fixed brain tissues were washed in phosphate-buffered saline (PBS) and transferred in 30% sucrose. Once the tissue was ready, 20 µm coronal cryosections were cut and sampled on glass slides. 

### 4.4. Immunohistochemistry

Coronal brain sections, including the striatum and SN, were selected for immunofluorescence staining. For immunofluorescence staining, sections were washed with 0.01 M phosphate buffer saline with 0.1% TritonX-100 (PBST, pH 7.4) and blocked in 5% normal horse serum (NHS) in PBST for 30 min at room temperature, followed by overnight incubation with the primary antibody at 4 °C. After incubation, sections were washed in PBS and incubated with secondary antibody cocktail VectaFluor™ Duet (Vector Laboratories, DK-8828, Newark, CA, USA) for an hour at room temperature. The sections were then washed in PBS and mounted with Vectashield mounting media with DAPI (Vector Laboratories, H-1200) [71]. Primary antibodies used for immunofluorescence staining were: marker proteins for DA neurons and fibers, tyrosine hydroxylase (TH, Abcam ab 113, Waltham, MA, USA), a marker protein for α-syn phosphorylated 129 (p-α-syn, Cell Signaling 23706S, Danvers, MA, USA), a marker for microglia, ionized calcium-binding adaptor protein-1 (Iba1, Abcam ab153696), glial fibrillary acidic protein (GFAP, InVitrogen, 14-9892-82, Waltham, MA), calpain-1 (Cell Signaling Technology, 2556S), calpain-2 (Cell Signaling, 2539), and arginase1 (Santa Cruz Biotechnology, Inc., sc-166920, Dallas, TX, USA). Images were evaluated and captured with an Olympus IX73 microscope. 

ImageJ software [71], and cellSens Imaging Software (OLYMPUS, Waltham, MA, USA) ere used for quantification of fluorescence images and counting cells. Three to five sections from each sample were used for quantification, and the sections were nearly 160–240 µm apart. The TH fiber density was measured in SNpr by ImageJ. At least three sections from each sample were used for this type of quantification. To analyze the integrated density, the image was converted into an 8-bit type and then into a binary image. A threshold was set on the binary image. On the dorsal striatum, an area was first selected and then integrated density was measured. 

### 4.5. Western Blot Analysis

Brain samples were homogenized in a standard homogenizing buffer (10 mM Tris-HCl + 150 mM NaCl, pH 7.4 + 1% Triton-X 100) with protease and phosphate inhibitor cocktail (ThermoFisher Scientific, 78440; Waltham, MA, USA) on ice. Protein was measured using the colorimetric assay based on the Lowry assay using Bio-Rad Protein Assay Kit (DC^TM^ Protein Assay Reagents, Hercules, CA, USA). An amount of 20 µg of proteins were loaded and electrophoresed on a 4–12% Bis/Tris NuPage gel (Invitrogen, Grand Island, NY, USA) [72,73]. Subsequently, the separated proteins were transferred onto a nitrocellulose membrane (Pierce, Rockford, IL). The blot was probed with calpain-1 (1:600, Cell Signaling, 2556), calpain-2 (1:500, Cell Signaling, 2539), and α-synuclein-phosphorylated 129 (p-α-syn, 1:500, Cell Signaling 23706S) antibodies. As a protein loading control, the monoclonal antibodies for GAPDH (1:500, Santa Cruz, sc-47724) and β-actin (1:1000, Santa Cruz, sc-47778) were used. The respective secondary antibodies consisting of horseradish peroxidase-conjugated anti-mouse (1:1000, Santa Cruz, sc-2005) and anti-rabbit (1:2000, Santa Cruz, sc-2004) were used. Blots were incubated with ECL detection reagents (Amersham Pharmacia, Buckinghamshire, UK), and pictures were taken using an Azure Biosystems c600 Imager. 

Using ImageJ software (National Institutes of Health, Bethesda, MD, USA), protein expression for each sample was quantified and expressed as relative density [74]. Relative protein density signifies the ratio of the expression for the protein of interest to the GAPDH expressed for each sample. 

### 4.6. Statistics

Statistical analyses were performed using Microsoft Excel and GraphPad Prism (version 6.0) Software. The immunoreactive bands obtained from Western blotting and the immunoreactive pixels of the immunofluorescence data were analyzed with ImageJ software (U.S. National Institutes of Health, Bethesda, MD, USA). A two-tailed paired t-test and a one-way ANOVA with Bonferroni test for multiple comparisons were used to determine statistical significance for all other analyses. Data were expressed as mean ± SEM or mean ±/STDEV. A *p*-value < 0.05 was determined to be statistically significant for all calculations.

## Figures and Tables

**Figure 1 ijms-23-13849-f001:**
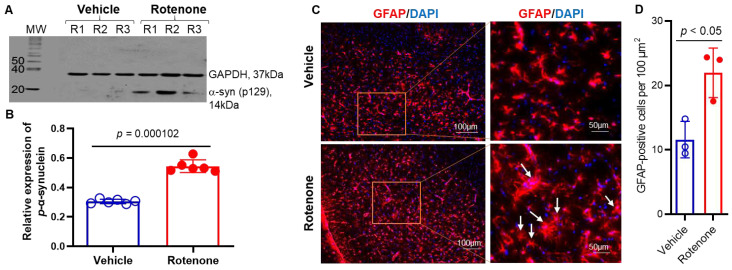
Administration of rotenone-induced increased expression of p-α syn 129 in the dorsal striatum of Lewis rats. (**A**) A representative photomicrograph of Western blot analysis showed increased expression of p-α-syn 129 in the striatum. GAPDH was used as a loading control. MW = molecular weight in kDa. (**B**) Densitometric analysis of p-α-syn 129 protein expression by ImageJ software shows that there is a significant increase (*p* < 0.001) in the synuclein protein expression in rotenone rats. N = 6. (**C**) Presence of highly active GFAP (red)-immunostained astrocytes in the SN following rotenone administration in rats. DAPI (blue) was used for nuclear staining. Representative microphotographs from control and rotenone groups suggest GFAP-immunostained astrocytes are hypertrophied in appearance in rotenone-injected rats as compared to vehicle controls. (**D**) Quantitative analysis of GFAP+ cells by cellSens Imaging Software suggests that astrocyte numbers were significantly increased following rotenone injection. N = 3.

**Figure 2 ijms-23-13849-f002:**
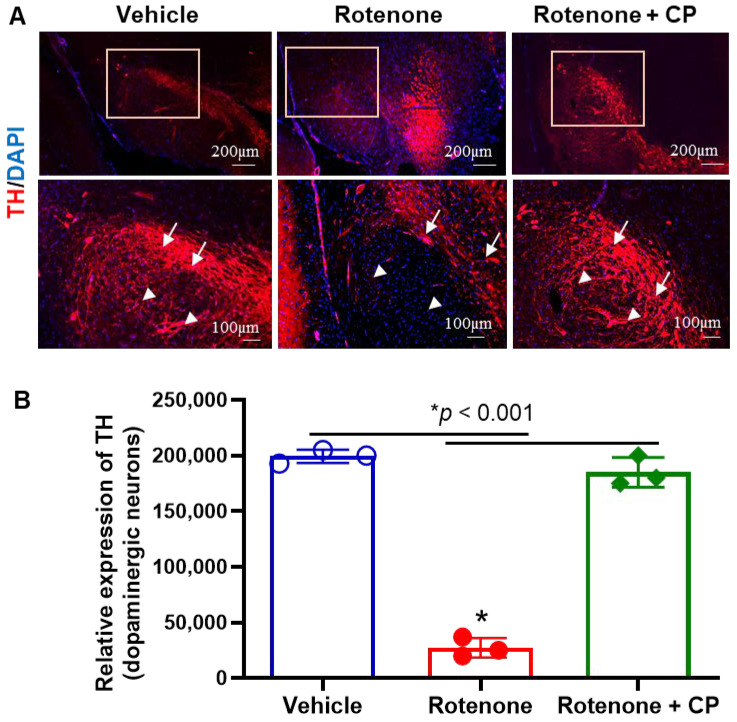
Rotenone-induced distinct loss of SN DA neurons and fibers. (**A**) Immunohistochemistry of rat SN samples with TH (red) following rotenone administration and calpain inhibition by CP treatment. Distinct loss of TH-positive neurons and fibers were seen in the rotenone rats as compared to controls. DAPI (blue) was used for nuclear staining. Note the dense TH staining of neurons in the SNpc (arrows) and DA fibers in SNpr (arrowheads) in the control group and also in rotenone plus CP treatment group. The extensive loss of TH-labeled neurons (arrows) and fibers (arrowheads) was demonstrated after rotenone administration. However, calpeptin treatment (rotenone+ CP) prevented neuronal loss in SN. (**B**) Analysis of TH-immunostained fiber density in SNpr by ImageJ software shows that there was a significant decrease (* *p* < 0.001) in TH fiber density in rotenone rats, which was prevented by CP treatment. N = 3.

**Figure 3 ijms-23-13849-f003:**
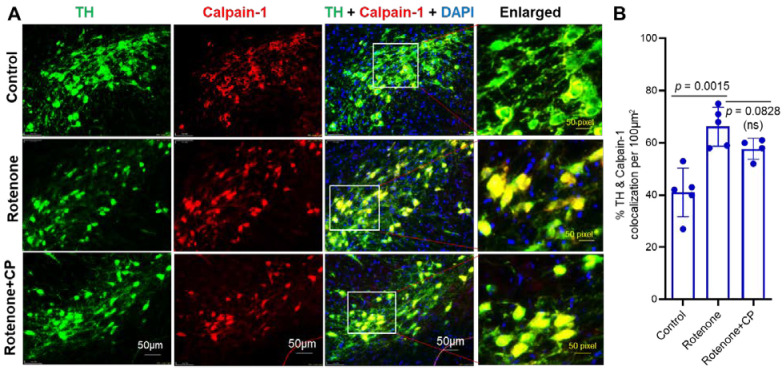
Calpain-1 expression was upregulated in SN DA neurons following rotenone administration, and it was retained after CP treatment. (**A**) Representative images from the respective treatment groups showed that TH-immunostained SN DA neurons co-localized with calpain-1 (arrows). Calpain-1 staining (red) indicates cytoplasmic localization in the SN neurons stained with TH (green). Merged images showed co-localization (yellow) of TH+ calpain-1 with a distinctly DAPI counter-stained nucleus (blue/purple color) after CP treatment. N = 3. (**B**) Quantitative analysis of TH+ and calpain-1+ cells colocalized per 100 µm^2^ following CP treatment. N = 4–5.

**Figure 4 ijms-23-13849-f004:**
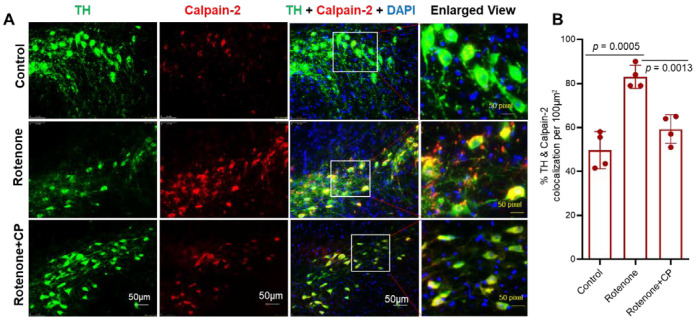
Calpain-2 expression was upregulated in SN DA neurons following rotenone administration, and it was not retained after CP treatment. (**A**) Representative images from the respective treatment groups showed SN DA neurons immunostained with TH (green) and calpain-2 (red). Co-localization of TH and calpain-2 (merged images) with DAPI (blue/purple) as a counter stain. Merged images showed less co-localization (yellow) of TH+ calpain-2 with a distinctly DAPI counter-stained nucleus (blue/purple color) after CP treatment. N = 3. (**B**) Quantitative analysis of TH+ and calpain-2+ cells colocalized per 100 µm^2^ following CP treatment. N = 4.

**Figure 5 ijms-23-13849-f005:**
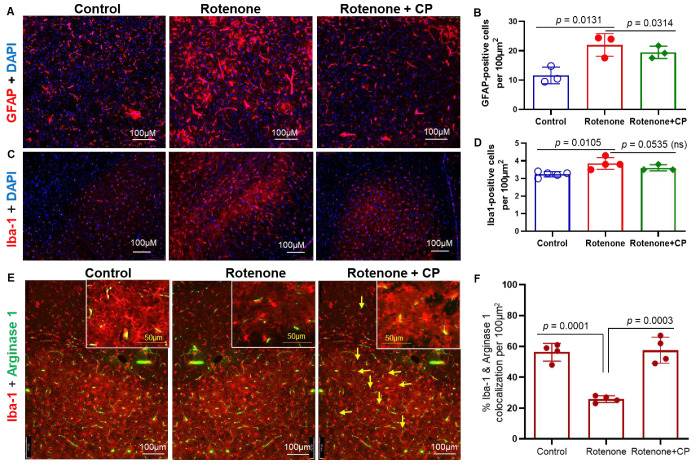
Activated astrocytes in the rat dorsal striatum following rotenone administration. Representative photomicrographs demonstrate GFAP-labeled astrocytes in the striatum in control, rotenone, and rotenone + CP treatment groups. (**A**) Rotenone administration conspicuously increased the number and size of the astrocytes in the striatum, and treatment of rats with CP decreased the number of reactive astrocytes. (**B**) Quantitative analysis by cellSens Imaging Software showed that the number of reactive astrocytes was significantly increased (*p* = 0.0131) in rotenone vs. controls, whereas it was markedly decreased (*p* = 0.0314) following CP treatment. N = 3. (**C**,**D**) Immunohistochemistry with Iba1 showed that the microglial population was significantly increased (*p* = 0.0105) in the SN after rotenone administration of rats, which were not significantly changed (*p* = 0.0535) following CP treatment. (**E**) Differentiation of microglia in rotenone rats after CP treatment. Colocalization (yellow, arrows) of Iba1 (red) and arginase 1 (green) in the dorsal striatum of CP-treated rats indicated differentiation of microglia into M2-type following calpain inhibition. N = 3–5. (**F**) Quantitative analyses of co-localized Iba-1+- and Arginase 1+-stained cells from Figure 5E. N = 4.

**Figure 6 ijms-23-13849-f006:**
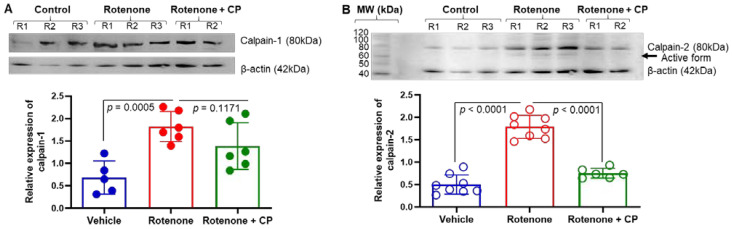
Administration of rotenone differentially influenced the expression levels of calpain-1 and calpain-2 in the nigrostriatal pathway in rats. (**A**) A representative photomicrograph of Western blot analysis showed upregulation of calpain-1 protein expression in the striatum following rotenone injection in rats (upper panel). β-actin was used as a loading control. Quantification of the protein level detected in Western blot analysis suggests calpain-1 expression was significantly increased (*p* = 0.0005) after rotenone injection, and it was marginally decreased (*p* = 0.1171) following CP treatment (lower panel, N = 5–6). (**B**) A representative photomicrograph of Western blot analysis also showed upregulation of calpain-2 protein expression in the striatum following rotenone injection (upper panel). β-actin was used as a loading control. Quantification of the calpain-2 protein level suggests a significant increase (*p* < 0.0001) in calpain-2 protein expression in the rotenone rats as compared to control group (lower panel, N = 6–8). CP treatment significantly decreased calpain-2 protein expression (*p* < 0.0001) in rotenone rats. MW = molecular weight.

**Figure 7 ijms-23-13849-f007:**
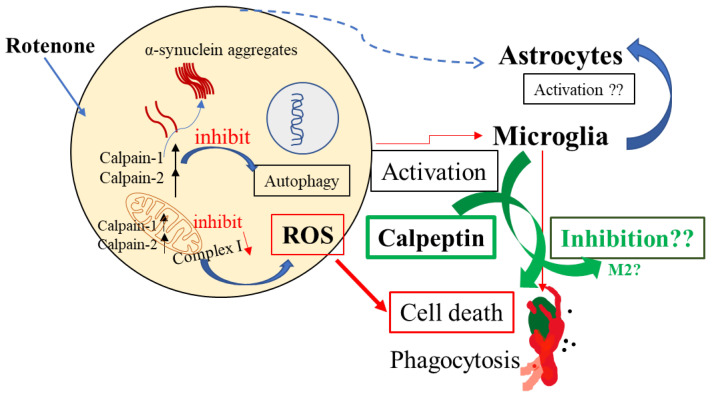
Schematic presentation showing rotenone-induced toxicity in SN neurons. Reactive astrocytes/microglia were detected after rotenone administration in Lewis rats. Both calpain-1 and calpain-2 were also activated in the nigrostriatal pathway following rotenone administration in rats. Calpain inhibition may help reduce glial activation via reduction of ROS and differentiation of microglia into M2-type cells. Decreased phagocytic function of glia and selective autophagy may also support preferential regulation of calpain-1 and calpain-2 expression/activity and the fate of neuronal survival in PD.

## Data Availability

The data used to support the findings of this manuscript are available from the corresponding authors upon reasonable written request.

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
