# Peer review of "Inhibition of Calpain Attenuates Degeneration of Substantia Nigra Neurons in the Rotenone Rat Model of Parkinson’s Disease"

_ijms, 2022, doi:10.3390/ijms232213849_

Round 1

Reviewer 1 Report

In the manuscript “Inhibition of calpain attenuates degeneration of substantia nigra neurons in the rotenone rat model of Parkinson’s Disease,” the authors evaluate a pan-calpain inhibitor, calpeptin, as a therapeutic target for rotenone induced parkinsonism. The authors establish their rotenone dosing as valid and then assess the effect of calpain inhibition on dopaminergic neuron loss, gliosis, and reactive oxygen species accumulation. They assert based on their findings that inhibition of calpain protects against many of the major pathologies associated with rotenone parkinsonism.

Overall, the authors present interesting ideas into the protective role of calpeptin as a therapeutic target for rotenone parkinsonism. However, there are several majors flaws with this manuscript in its current form. The most concerning of these flaws is the lack of quantification on the immunofluorescence images. Several assertions are made by the authors that a change in expression, colocalization, or morphology is indicated in the images. It is impossible to ascertain whether these changes are occurring without including some form of quantification. In addition to this, the quality of images included in the figures make morphological assessment very difficult. A much greater image resolution is needed in order to properly assess morphology in both astrocytes and microglia. Another major imaging concern is the lack of intensity standardization. The intensity of the images is considerably variable making it difficult to assess differences in Calpain 1/2 colocalization with TH in the different animal groups. Another concern is the assertion that different Calpain isoforms play opposite roles in degeneration. This claim is not validated by any experiments nor are there sufficient citations to make this assertion. My final major concern is with the statistical reporting. In several places the N reported does not match up with the number of data points in the graphs (Figures 2, 5, and 7), and in Figure 7, the N reported in the graph is not sufficient to run data analysis on. This is concerning because significance is reported despite an insufficient N, and that calls into question the rest of the statistical analyses.

Minor Comments:

1.    The inclusion of detailed summarized results from a previous study beginning on line 72 are unnecessary and confusing. While these results are interesting and should be included in the discussion of treatment with calpeptin, the current position detracts from the results of the current study.

2.    In several places throughout the manuscript text appears to be in a different font or font size. Standardize throughout the document. Examples: Line 111 “while calcium”. Line 112-113 “calcium induces calpain activation and”. Line 159 “Astrocytes”.

3.    In section 2.2 of the results, the discussion of the dorsal striatum in the first sentence is out of place. While the information is true, consider moving it to a section where the striatum itself is being investigated.

4.    In lines 151-153, the two sentences are redundant. Consider removing one or rewriting to make the point clearer.

5.    Section 2.6 of the results evaluates ROS following the treatment of rotenone in SH-SY5Y cells. It is unclear why the authors chose to move backwards into an in vitro line when they have utilized an animal model up to this point in the manuscript. The authors also chose to evaluate a different inhibitor than the primary subject of this paper. This section is not cohesive with the rest of the manuscript. Consider evaluating ROS in tissue to compliment existing data.

Author Response

Manuscript ID: ijms-1882398

Manuscript title: Inhibition of Calpain Attenuates Degeneration of Substantia Nigra Neurons in the Rotenone Rat Model of Parkinson’s Disease

Reviewer #1

We appreciate the reviewer’s comment “Overall, the authors present interesting ideas into the protective role of calpeptin as a therapeutic target for rotenone parkinsonism.” However, the reviewer raised some concerns which are addressed below:

Major concerns:

Concern #1:  “The most concerning of these flaws is the lack of quantification on the immunofluorescence images. Several assertions are made by the authors that a change in expression, colocalization, or morphology is indicated in the images.”

Response #1:  We appreciate the reviewer’s comments on the quantification of the immunofluorescence images. Quantitative analyses of appropriate images are now included in the revised manuscript (Figures 1, 2, 5, and 6).

Concern #2:  “The quality of images included in the figures make morphological assessment very difficult. A much greater image resolution is needed in order to properly assess morphology in both astrocytes and microglia. Another major imaging concern is the lack of intensity standardization. The intensity of the images is considerably variable making it difficult to assess differences in Calpain 1/2 colocalization with TH in the different animal groups.”

Response #2:  We agree with the reviewer that a much greater image resolution is needed in order to properly assess morphology in both astrocytes and microglia. Thus, a greater image resolution is included and the intensity was adjusted and normalized to properly assess morphology in both astrocytes and microglia. This revised standardized intensity of the images should be helpful to assess differences in Calpain 1/2 colocalization with TH in the different animal groups and was further discussed in the revised text.

Concern #3:  “My final major concern is with the statistical reporting. In several places the N reported does not match up with the number of data points in the graphs (Figures 2, 5, and 7), and in Figure 7, the N reported in the graph is not sufficient to run data analysis on. This is concerning because significance is reported despite an insufficient N, and that calls into question the rest of the statistical analyses.”

Response #3:  We have carefully looked at the statistical reporting and revised the number of data points in the graphs (Figures 2 and 5, and new revised figure 6).  Furthermore, the N reported in the graph was increased including adding the results from 2-4 additional samples in Figure 7 (revised Figure 6), and quantitative analyses were performed using Two-tailed paired t-test and one-way ANOVA with Bonferroni test for multiple comparisons to determine statistical significance for all analyses. A revised Figure 6 is also included and this was further discussed in the revised text.

Minor comments:

Comment #1:  “The inclusion of detailed summarized results from a previous study beginning on line 72 are unnecessary and confusing. While these results are interesting and should be included in the discussion of treatment with calpeptin, the current position detracts from the results of the current study.”

Response #1:  We agree with the reviewer that the inclusion of detailed summarized results from a previous study beginning on line 72 are unnecessary and confusing, and thus it was deleted from the revised text. Our current results were also discussed in light of the previous and current findings as suggested.

Comments #2:  “In several places throughout the manuscript text appears to be in a different font or font size. Standardize throughout the document. Examples: Line 111 “while calcium”. Line 112-113 “calcium induces calpain activation and”. Line 159 “Astrocytes”.

Response#2:  We apologize for unwanted alterations of the font size. It is now standardized throughout the document.

Comment #3:  “In section 2.2 of the results, the discussion of the dorsal striatum in the first sentence is out of place. While the information is true, consider moving it to a section where the striatum itself is being investigated.”

Response #3:  We agree with the reviewer that the discussion of the dorsal striatum in the first sentence in section 2.2 is not needed. Thus, this sentence was deleted from the revised text..

Comment #4:  “In lines 151-153, the two sentences are redundant. Consider removing one or rewriting to make the point clearer.”

Response #4:  We have removed one redundant sentence from the revised text (Section 2.3) as suggested.

Comment #5:  “Section 2.6 of the results evaluates ROS following the treatment of rotenone in SH-SY5Y cells. It is unclear why the authors chose to move backwards into an in vitro line when they have utilized an animal model up to this point in the manuscript. The authors also chose to evaluate a different inhibitor than the primary subject of this paper. This section is not cohesive with the rest of the manuscript. Consider evaluating ROS in tissue to complement existing data.”

Response #5:  We appreciate the reviewer’s insightful comments on ROS regarding rotenone treatment of SH-SY5Y cells. We also agree that this section 2.6 moves backwards into an in vitro line up to this point in the manuscript. We have previously shown that oxidative stress induces calpain activation and that calpain inhibition decreases ROS production in a number of in vitro and in vivo settings (Brain Res 2000, 852(2):326-34; J Neurochem 2014, 130:280-90; J Neurochem 2016, 139:440-455; Neurotox Res 2020, 38:640-649). Thus, we have deleted Figure 6 from the revised manuscript because it would be redundant to add it to the current manuscript.

Reviewer 2 Report

Authors investigated the differential role for calpain 1 and 2 in rotenone-induced PD rat models. 

My major concern:

- I do not see any clear evidence  showing two activated form of calpains plays an opposite role in their rat PD model.   Do they check this conclusion with more specific calpain 1 or calpain 2 inhibitors ? I do not see what other criteria than IHC are used before they make a such conclusion. 

- For fig.7, more animals should be used to make a clear conclusion of regulation of calpain 2 expression pattern in three groups.

Other comments and suggestions

- In Fig 2, authors should describe whether stereological counting and densitometric measurement of fiber density are conducted.

- in Fig 1 and 2, lower magnification images showing the whole striatum and midbrain seem to be informative to judge the damaged areas of rat PD model

- In other figures, enlarged merged view is also required to clearly judge the cytoplasmic vs nuclear localization.

- In fig.1 and 7, size marker must be indicated at the side of immunoblot.

- In fig.1, bar size of image is missing.

Author Response

Manuscript ID: ijms-1882398

Manuscript title: Inhibition of Calpain Attenuates Degeneration of Substantia Nigra Neurons in the Rotenone Rat Model of Parkinson’s Disease

Reviewer #2

Major concerns:

Concern #1:  “I do not see any clear evidence showing two activated form of calpains plays an opposite role in their rat PD model.   Do they check this conclusion with more specific calpain 1 or calpain 2 inhibitors? I do not see what other criteria than IHC are used before they make a such conclusion.” 

Response #1:  We understand the reviewer’s concern about the questions related to more specific calpain 1 or calpain 2 inhibitors. Unfortunately, there is no specific calpain-1 inhibitor available at this time that can be used to test in animal models. However, studies using calpain-1 knockout and calpain-2 conditional knockout mice have shown that calpain-1 could be neuroprotective and calpain-2 may lead to neuronal death or degeneration (Cells 2020, 9, 12; Neurotrauma 2018, 35:105-117).  Our study suggests that the expression of calpain-1 remains unchanged or marginally reduced following pan calpain inhibitor treatment in rotenone rats (Figure 6A). However, calpain-2 was significantly inhibited (Figure 6B) which correlated with improved outcome in rotenone rats, suggesting that calpain-2 inhibition could be neuroprotective. Thus, based on the current literature on the regulation of cell survival and cell death, respectively, by calpain-1 and calpain-2, and our present study on differential expression of calpain-1 and calpain-2 in neurodegenerative events in rotenone rats suggested that they may play opposing roles in neurodegenerative diseases. This information is now included in the revised manuscript.

Concern #2:  “For fig.7, more animals should be used to make a clear conclusion of regulation of calpain 2 expression pattern in three groups.”

Response #2:  We agree with the reviewer that more animals should be used to make a clear conclusion of regulation of calpain 2 expression pattern in three groups. Thus, we have increased the number of animals in the western blot analysis of samples.  A revised bar graph (Figure 6B) with additional samples and statistical analyses were included in the revised manuscript.

Other comments and suggestions:

Comment #1:  “In Fig 2, authors should describe whether stereological counting and densitometric measurement of fiber density are conducted.”

Response #1:  The reviewer is right. Stereological counting and densitometric measurement of fiber density were conducted and presented in the figure 2.

Comment #2:  “In Fig 1 and 2, lower magnification images showing the whole striatum and midbrain seem to be informative to judge the damaged areas of rat PD model”.

Response #2:  We agree with the reviewer and lower magnification images for Figures 1 and 2 were included in the revised manuscript showing the whole striatum and midbrain. These new data should be informative as suggested by the reviewer.

Comment #3:  “In other figures, enlarged merged view is also required to clearly judge the cytoplasmic vs nuclear localization.”

Response #3:  According, enlarged view of the images for calpain1- and calpain-2 co-localization are also added (Figures 3, 4, and 5) in the revised manuscript.

Comment #4:  “In fig.1 and 7, size marker must be indicated at the side of immunoblot.

- In fig.1, bar size of image is missing.”

Response #4:  We agree with the reviewer that the size marker must be indicated at the side of immunoblot. Thus, in Figure 7 (new figure 6) molecular weight markers are included at the side of the blot indicating the size of the protein. In addition, the bar size is also included in Figure 1.

Reviewer 3 Report

Zaman et al. have demonstrated the effect of calpeptin on calpain1 and 2 in the substantia nigra of rotenone-induced rat model for PD. Below are the limitations detected in the submitted manuscript:

Major:

1)    There are multiple sentences in the introduction and discussion where the authors state clear cause-effect relationships, which have not yet been clearly established. The authors need to clarify the language in the text in order to express published versus hypothetical relationships.

For example, in line 59, they say the a-syn fragments induce oxidative stress which leads to DA neuron death. This has not been established yet.

Example, line 68-70.

Example. Line 99-101.

2)    The authors state that calpain-1 levels were unaffected with Calpeptin treatment in Rotenone-treated rats. And this means that calpain-1 is required for survival of DA TH neurons in presence of rotenone. This is indecipherable with the data provided. The shown data only tells us that calpain-1 is not critical for TH neuron survival, since its levels are not impacted by Calpeptin, although the survival of these neurons are increased by CP. The authors need to change these statements were there is clear over-interpretation of the data provided.

3)    Adding more on the above point, the shown data does not in any way suggest that calpain-1 and -2 have opposite effects on TH neuron survival. The authors are advised strongly to change these unsupported statements in the manuscript.

Minor:

1)    Include reference for the TH being a substrate for calpains in line 49-50 of introduction.

2)    Please explain the rotenone rat model in short at the beginning of the Results section. This will help the readers.

3)    Include reference for “While calcium homeostasis is important for the maintenance of neuronal integrity, elevated calcium induces calpain activation and aggregation of -syn, and stimulation of gliosis in PD.”

4)    Please clarify this statement further “Calpain-1 immunofluorescence staining in the SN of rotenone rats showed precise localization of this protease in SN neurons and fibers in the control rats”

5)    Please provide reference for “Astrocytes possess a calcium-based form of excitability “

6)    Please provide a line or two explaining Iba and arginase expression in relation to microglia in Results section 2.5

7)    Explain the link between reactive astrocytes and neuroinflammation in section 2.5

8)    There is major rewording required in Results section 2.6: The data shows that CP affects the levels of calpain-2. We don’t know if it impacts the expression of calpains, since the experiment does not involve cycloheximide or degradation pathway inhibitors. Hence, CP can only either increase or decrease the calpain levels, and not inhibit their expression, based on given data.

9)    “Our study however found enhanced expression of both calpain isoforms in SN DA neurons, suggesting calpain expression may be related to SN neuroinflammation rather than local DA cell death “. This line in Discussion is over-interpretation again. These events are merely observed in the tissues, there is no cause-effect or relation shown here.

10) TH expression alone can be decreased in a PD model, without neuronal death. Since this is a possibility which can influence their results, the authors need to mention this in the manuscript.

11) The authors also need to address the limitations of the rotenone-induced PD model in the Discussion. This will provide the whole picture to the readers.

Author Response

We appreciate the third reviewer’s time and comments regarding our manuscript (ijms-1882398) on the effect of calpeptin on calpain-1 and calpain-2 in the substantia nigra of rotenone-induced rat model for PD. Below are point by point responses to the comments. Changes in the revised text are highlighted in blue.

            Major:

Comment #1: “There are multiple sentences in the introduction and discussion where the authors state clear cause-effect relationships, which have not yet been clearly established. The authors need to clarify the language in the text in order to express published versus hypothetical relationships.

For example, in line 59, they say the a-syn fragments induce oxidative stress which leads to DA neuron death. This has not been established yet.

Example, line 68-70. Example. Line 99-101.”

Response #1:  We appreciate the reviewer’s comments and agree that there are limitations of this study. However, we have reservations about “the a-syn fragments induce oxidative stress which leads to DA neuron death.” Lines 59, 68-70, and 99-101 are now revised, as suggested by the reviewer. A reference regarding the involvement of a-syn fragments in oxidative stress and cell death is also included (Cellular and Molecular Life Sciences 2008, 65:1272-1284). In addition, the limitations of this study are now discussed in a separate paragraph at the end of the discussion section of the revised manuscript.

Comment #2: “The authors state that calpain-1 levels were unaffected with Calpeptin treatment in Rotenone-treated rats. And this means that calpain-1 is required for survival of DA TH neurons in presence of rotenone. This is indecipherable with the data provided. The shown data only tells us that calpain-1 is not critical for TH neuron survival, since its levels are not impacted by Calpeptin, although the survival of these neurons are increased by CP. The authors need to change these statements were there is clear over-interpretation of the data provided.”

Response #2:  Our data suggested that calpain-2 was significantly inhibited by calpeptin while calpain-1 inhibition was marginal and not statistically significant. These data were included and discussed in the manuscript, suggesting that calpain-1 may be necessary for neuronal survival while calpain-2 may be neurodegenerative.  Other laboratories have also reported this in different preclinical studies (please see discussion, yellow highlights). However, we agree that some of the statements made in the manuscript should be revised to avoid over-interpretation of the data provided. Thus, these statements are revised, and limitations of the study are now included, as suggested.

Comment #3: “Adding more on the above point, the shown data does not in any way suggest that calpain-1 and -2 have opposite effects on TH neuron survival. The authors are advised strongly to change these unsupported statements in the manuscript.”

Response #3: We understand the reviewer’s concerns. We are not strictly suggesting calpain-1 and -2 have opposite effects on TH neuron survival. Our data suggests that overactivation of calpain is detrimental for neuronal survival. While both calpain-1 and calpain-2 are essential enzymes for cellular function, overactivation of calpain-2 could be more degenerative than calpain-1. Studies from other laboratories also support the hypothesis that calpain-1 is required for neuronal survival in preclinical models (please see discussion). However, some of these strictly unsupported statements in the manuscript are now revised.

Minor:

Comment #1: “Include reference for the TH being a substrate for calpains in line 49-50 of introduction.”

Response #1: A reference for the TH being a substrate for calpains is now included, as recommended (Mol Neurobiol 2008, 38: 78-100).

Comment #2: “Please explain the rotenone rat model in short at the beginning of the Results section. This will help the readers.”

Response #2: We agree that explaining the rotenone rat model in short at the beginning of the Results section will help the readers. Thus, the rotenone rat model was discussed in short at the beginning of the Results section as suggested. The following statements are included at the beginning of the Results section in the revised manuscript:

“Rotenone is a naturally occurring compound and is used as a piscicide.  It is a potent inhibitor of complex-1 of mitochondrial respiratory chain (Scientific Reports 2017, 7: 45465). The inhibition of complex-I causes highly selective nigrostriatal degeneration in animal models (Nat Neurosci 2000, 3:1301-6; Neurobiol Dis 2009, 34: 279-90). Moreover, this model also provides evidence that the mitochondria play a critical role in the neurodegenerative disease process (Parkinsonism Relat Disord 2003, 9 Suppl 2: S59-64). In the present study, rotenone is used to induce Parkinson-like disease in the rat model to investigate the roles of two major calpain isoforms (calpain-1 and calpain-2) in the pathogenesis of Parkinson’s disease.”

Comment #3: “Include reference for “While calcium homeostasis is important for the maintenance of neuronal integrity, elevated calcium induces calpain activation and aggregation of a-syn, and stimulation of gliosis in PD.”

Response #3: References are now included to support the statement “calpain activation and aggregation of a-syn, and stimulation of gliosis in PD” in the revised manuscript, as suggested (The International Journal of Biochemistry & Cell Biology 2002, 34:722-725; Biochemistry 2005, 44:7818–7829;).

Comment #4: Please clarify this statement further “Calpain-1 immunofluorescence staining in the SN of rotenone rats showed precise localization of this protease in SN neurons and fibers in the control rats”

Response #4: The statement “Calpain-1 immunofluorescence staining in the SN of rotenone rats showed precise localization of this protease in SN neurons and fibers in the control rats” is now clarified, as suggested. The following statements are included in the revised manuscript: “As shown in the Figure 3, the SN neurons and fibers immunostained specifically with TH showing that these nigral neurons are dopaminergic neurons. The co-localization of these specific TH-stained neurons with calpain-1 indicates that the calpain-1 expressed specifically in the SN dopaminergic neurons and fibers.”

Comment #5: “Please provide reference for “Astrocytes possess a calcium-based form of excitability.”

Response #5: Three references are provided to support the statement “astrocytes possess a calcium-based form of excitability.” (Nature Neuroscience 2001, 4:803–812; Methods Mol Biol 2009, 489:93-109; Sci Rep 2020, 10:14474).

Comment #6: “Please provide a line or two explaining Iba and arginase expression in relation to microglia in Results section 2.5.”

Response #6: We appreciate the reviewer’s comment. The following sentences are added explaining Iba-1 and arginase 1 expression in relation to microglia in revised Results section 2.5.

“Neuroinflammation in the brain can be detected by the presence of activated microglia. The marker Iba-1 immunostaining is used to detect microglia in SN and striatum.  The phenotypes of activated microglia are broadly classified into two types based on their proinflammatory (M1) and anti-inflammatory (M2) responses (J Neuroimmune Pharmacol 2009, 4:399-418; Nat Rev Immunol 2003, 3:23-35). Arginase-1 is one of the best characterized markers for M2 type microglia (J Immunol 1999, 163:3771-3777; J Neuroinflammation. 2014; 11: 98), and the co-localization of Iba-1 +arginase-1 immunostaining was used to classify the microglial phenotypes following treatments. This co-localization of marker proteins indicated if calpain inhibition altered microglial phenotype. The presence of arginase-1 positive microglia in rotenone+calpeptin group compared to rotenone treatment group suggests that type 2 (M2) microglia may be helping in the repair process in the brain after calpain inhibition.”

Comment #7: “Explain the link between reactive astrocytes and neuroinflammation in section 2.5.”

Response #7: The link between reactive astrocytes and neuroinflammation was explained in section 2.5, as suggested. The following sentences are included in the revised manuscript.

“Neuroinflammation in the SN is a consistent feature of PD (Brain 2013, 36:2419–2431). Reactive astrocytes play a complex role under different pathological situations.  These astrocytes can contribute to the disruption of blood-brain barrier and inflammatory events in patients with PD (J. Cereb. Blood Flow Metab 2015, 35:747-750).

Comment #8: “There is major rewording required in Results section 2.6: The data shows that CP affects the levels of calpain-2. We don’t know if it impacts the expression of calpains, since the experiment does not involve cycloheximide or degradation pathway inhibitors. Hence, CP can only either increase or decrease the calpain levels, and not inhibit their expression, based on given data.”

Response #8: We agree with the reviewer (with some reservation) that whether CP strictly impacts the expression of calpains. Our study is not focused on protein synthesis machinery where the experiment may involve cycloheximide or degradation pathway inhibitors. We have been working on calpain and calpain inhibitors for more than 30 years using many different neurodegenerative disease models where CP impacted the expression of calpains. Thus, we respectfully disagree with the comment that CP can only either increase or decrease the calpain levels, and not inhibit their expression, based on given western blot data. However, our western blot results also suggest CP treatment alters the calpain levels. Thus, these statements have been revised and included in the revised text as suggested.

Comment #9: “Our study however found enhanced expression of both calpain isoforms in SN DA neurons, suggesting calpain expression may be related to SN neuroinflammation rather than local DA cell death “. This line in Discussion is over-interpretation again. These events are merely observed in the tissues, there is no cause-effect or relation shown here.”

Response #9: Since the reviewer thinks the line in discussion “Our study however found enhanced expression of both calpain isoforms in SN DA neurons, suggesting calpain expression may be related to SN neuroinflammation rather than local DA cell death” is over-interpretation, it is now removed from the revised text.

Comment #10: “TH expression alone can be decreased in a PD model, without neuronal death. Since this is a possibility which can influence their results, the authors need to mention this in the manuscript.”

Response #10: We agree that TH expression alone can be decreased in a PD model without neuronal death. This is now mentioned in the revised text (discussion), as suggested.

Comment #11: “The authors also need to address the limitations of the rotenone-induced PD model in the Discussion. This will provide the whole picture to the readers.”

Response #11: We appreciate the reviewer’s insightful comment. Thus, the limitations and strengths of the rotenone-induced PD model were discussed at the end of the Discussion section. The following paragraph is included.

The limitations and strengths of the rotenone-induced PD model:  The investigation of neurodegenerative diseases is challenging.  While animal models can exhibit the same pathophysiological features as human disease, they do not exactly mimic human neurological disorders.  Thus, the rotenone model used here has some limitations and strengths.   Rotenone induced PD in rats is a popular model for the disease because it selectively degenerates nigrostriatal pathway. Most importantly, this model induces α-synuclein aggregation (Lewy body formation). Daily intraperitoneal administration of rotenone causes striatal dopamine depletion, degeneration of nigrostriatal dopamine, and loss of TH-positive neurons of the SN. Remarkably, the magnitude and distribution of the lesion is very similar across animals. Cytoplasmic inclusions containing α-synuclein in rats have morphology very similar to human Lewy bodies.  In addition, inflammation and oxidative damage make the rotenone model particularly attractive for studying disease-modifying therapies. The slow gastrointestinal motility due to rotenone administration is regarded as “severe digestive problem” (FASEB 2004, 18:717-9) and considered a limitation of this animal model. Chronic exposure to rotenone may cause the delayed appearance of parkinsonian α-synuclein pathology in the enteric nervous system and an associated functional deficit in gastrointestinal motility (Neurobiol Dis 2009, 36:96-102; Exp Neurol 2009, 218:154-61).  However, severe constipation and delayed gastric emptying are also prodromal symptoms of PD, as reported by most of the patients (Mov Disord 2019, 34:480-486).  The major limitation of the rotenone model is its variability, both in terms of the percentage of animals that develop a clear-cut nigrostriatal lesion and the extent of that lesion. However, the use of older animals produces the phenotype with much less temporal variability. The refinement of the treatment schedule also reduces the variability within a treatment group, and has been applied to this study (Neurobiol Dis 2009, 34:279-90). Efforts are being made to replicate familial PD by generating transgenic mice carrying distinct mutations. While they do not display clear dopaminergic neurodegeneration or parkinsonian motor deficits, they do show altered neuronal function and a-synuclein aggregation.  Overall, replicating neurological diseases in animal models remain challenging. 

Round 2

Reviewer 1 Report

The authors have revised a majority of comments raised during the first round of review. I have two remaining comments/concerns for the manuscript.

1.     A colocalization quantification analysis is needed to assess the overlap of stained proteins in figures 3, 4, and 5 E.

2.     Data points should be added to the graphs in figure 1 D.

Author Response

Manuscript ID: ijms-1882398

Manuscript title: Inhibition of Calpain Attenuates Degeneration of Substantia Nigra Neurons in the Rotenone Rat Model of Parkinson’s Disease

Reviewer #1, Round 2

We appreciate the reviewer’s remaining comments for the manuscript.

Comment #1: “A colocalization quantification analysis is needed to assess the overlap of stained proteins in figures 3, 4, and 5 E.”

Response #1: Quantitative analyses of stained proteins in colocalization study in Figures 3B, 4B, and 5F are now included in the revised manuscript.  Changes in the text are highlighted in green.

Comment #2: “Data points should be added to the graphs in figure 1 D.”

Response #2: We agree with the reviewer and the data points are now added to the graphs in Figure 1D.